# Efficacy, cost-minimization, and budget impact of a personalized discharge letter for basal cell carcinoma patients to reduce low-value follow-up care

Sven van Egmond[1☺]*, Ella D. van Vliet[2☺]*, Marlies Wakkee[1], Loes M. Hollestein[1,3], Xavier G. L. V. Pouwels[2], Hendrik Koffijberg[2], Yesim Misirli[3], Rachel S. L. A. Bakkum[4], Maarten T. Bastiaens[5], Nicole A. Kukutsch[6], Albert J. Oosting[7], Elsemieke I. Plasmeijer[8], Annik van Rengen[9], Kees-Peter de Roos[10], Tamar E. C. Nijsten[1], Esther de Vries[11], Esther W. de Bekker-Grob[12]

1 Department of Dermatology, Erasmus MC Cancer Institute, Rotterdam, The Netherlands, 2 Department of Health Technology and Services Research, University of Twente, Enschede, The Netherlands, 3 Department of Research, Netherlands Comprehensive Cancer Organization (IKNL), Utrecht, The Netherlands, 4 Department of Dermatology, Alrijne Ziekenhuis, Leiderdorp, The Netherlands, 5 Department of Dermatology, Elisabeth-TweeSteden Hospital, Tilburg, The Netherlands, 6 Department of Dermatology, Leiden University Medical Center, Leiden, The Netherlands, 7 Department of Dermatology, Spaarne Ziekenhuis, Hoofddorp, The Netherlands, 8 Department of Dermatology, The Netherlands Cancer Institute, Antoni van Leeuwenhoek, Amsterdam, The Netherlands, 9 Department of Dermatology, Mohs Klinieken, Dordrecht, The Netherlands, 10 Department of Dermatology, DermaPark, Uden, The Netherlands, 11 Department of Clinical Epidemiology and Biostatistics, Pontificia Universidad Javeriana, Bogota, Colombia, 12 Erasmus School of Health Policy & Management, Erasmus University, Rotterdam, The Netherlands

☺ These authors contributed equally to this work.
* s.vanegmond@erasmusmc.nl (SE); e.van.vliet95@gmail.com (EDV)

**Data Availability Statement:** All relevant data are within the manuscript and its Supporting Information files.

## Abstract

### Background

The incidence of keratinocyte carcinomas is high and rapidly growing. Approximately 80% of keratinocyte carcinomas consist of basal cell carcinomas (BCC) with 50% of these being considered as low-risk tumors. Nevertheless, 83% of the low-risk BCC patients were found to receive more follow-up care than recommended according to the Dutch BCC guideline, which is one visit post-treatment for this group. More efficient management could reduce unnecessary follow-up care and related costs.

### Objectives

To study the efficacy, cost-utility, and budget impact of a personalized discharge letter for low-risk BCC patients compared with usual care (no personalized letter).

### Methods

In a multi-center intervention study, a personalized discharge letter in addition to usual care was compared to usual care in first-time BCC patients. Model-based cost-utility and budget impact analyses were conducted, using individual patient data gathered via surveys. The

**Funding:** This project was funded by Citrienfonds (Dutch Ministry of Health, Welfare and Sport, https://www.citrienfonds.nl/, received by TN) and VGZ (Health insurance company, https://www.vgz.nl/, received by EdV). The funders were not involved in study design, data collection, data analysis, and manuscript preparation.

**Competing interests:** The authors have declared that no competing interests exist.

outcome measures were number of follow-up visits, costs and quality adjusted life years (QALY) per patient.

## Results

A total of 473 first-time BCC patients were recruited. The personalized discharge letter decreased the number of follow-up visits by 14.8% in the first year. The incremental costs after five years were -24.45 per patient. The QALYs were 4.12 after five years and very similar in both groups. The national budget impact was -2,7 million after five years.

## Conclusions

The distribution of a personalized discharge letter decreases the number of unnecessary follow-up visits and implementing the intervention in a large eligible population would results in substantial cost savings, contributing to restraining the growing BCC costs.

## Introduction

Keratinocyte carcinoma (KC) is the most common malignancy worldwide with still rising incidence rates [1–5]. Of these carcinomas, basal cell carcinoma (BCC) is by far the most common type, concerning 80% of all KCs in Caucasian populations. The remaining 20% consists of cutaneous squamous cell carcinoma (cSCC) [2, 6]. In 2017, over 48,000 individuals were newly diagnosed with a BCC in the Netherlands and this increases by 8% annually [7]. This alarming growth of new cases results in a strain on dermatological care and budget, with yearly costs for skin cancer alone already reaching 456 million euro [8, 9]. Total KC costs are now 1.8 times higher compared to melanoma [1, 3, 10]. In addition to focusing on skin cancer prevention, interventions aimed at improving efficiency of care, especially in the case of high-volume tumors such as the BCC, are equally essential to safeguard current care.

De-adoption of low-value care is a strategy that can be used to restrain costs. Low-value care is defined as "care that is unlikely to benefit a patient given the harms, costs, alternatives or preferences" [11]. After a low-risk BCC (i.e., primary BCC, < 2 cm, located outside the H-zone, with a nodular or superficial subtype), follow-up visits after the initial check-up can be labeled as low-value, because there is no evidence that extra follow-up provides a health benefit [12, 13]. According to the Dutch BCC guideline, annual follow-up should therefore be limited to high-risk patients only. Dermatologists are recommended to check the scar of low-risk patients just once within 6–12 months after treatment and are advised to instruct patients in self-examination and provide additional information via brochures [14]. About 50% of the BCC patients are considered low-risk, but research has shown that 83% of these low-risk patients receive more follow-up than the guidelines recommend during the first five years after treatment [3, 15]. There is currently no evidence that extra follow-up provides a health benefit. Therefore, care for low-risk BCC patients requires more efficient management and guideline adherence.

To avoid trial and error on de-adoption strategies, we conducted research on the needs and preferences of patients and dermatologists regarding current BCC management, integrated within a Choosing Wisely project [16]. BCC patients expressed the need for more information, tailored to their situation and indicated that this information would lower their need for frequent follow-up visits [17]. A discrete-choice experiment revealed that BCC patients accepted

fewer follow-up visits when provided a personalized printed discharge letter over other alternatives (e-health, general brochure or website) [18]. These letters can contain relevant information on a patient's diagnosis, treatment, complications, follow-up and lifestyle recommendations. Providing such a letter could reduce unnecessary follow-up visits among BCC patients and lower the costs of BCC management compared to current practice. The aim of this study was to explore the efficacy, cost-utility, and budget impact of a personalized discharge letter to first-time BCC patients in comparison with usual care.

## Patients and methods

The study was approved by the Medical Ethical Committee of the Erasmus MC (MEC-2014-374) and analyses were performed according to the (inter)national guidelines of cost-utility analysis (CUA) and a budget impact analysis (BIA), as well as the CHEERS checklist for reporting [19–23].

### Study population and design

The study population consisted of patients with a first BCC who were included after treatment but prior to follow-up. Patients needed to be at least 18 years old and had to be able to speak Dutch. Patients with a skin cancer diagnosis prior to their first BCC were excluded. Patients were included in six healthcare centers in the Netherlands; one academic hospital, three general hospitals, and two independent skin sector treatment centers. All participants were asked to complete a survey at baseline and after three, six, and twelve months. Each survey consisted of general questions regarding demographics, their quality of life (based on the EQ-5D-3L questionnaire, when this trial started there were no Dutch tariffs known for the 5L version), the number of BCC related visits to the general practitioner (GP) and medical specialist, whether they received any subsequent skin cancer diagnosis and the SF-HLQ questionnaire to monitor the effects of a BCC on labor activities [22, 24].

First, the control patients were included in 2014 and their data were collected via surveys. These patients received usual care and could be offered general dermatological brochures, whichever the dermatologists considered appropriate. A discrete choice experiment was conducted in this control group, which showed that a printed personalized discharge letter would reduce the need for unnecessary follow-up [25, 26]. This personalized discharge letter contained information on the patient's diagnosis, treatment, chance of having a subsequent BCC, lifestyle recommendations, information on self-examination and advice for actions when new suspicious lesions would appear (S1 Appendix in S1 File). In 2016, new BCC patients from the participating hospitals were included in the study, forming the intervention group. They received usual care with the personalized discharge letter post-treatment.

### Efficacy

The efficacy of the intervention was expressed as the percentage reduction in BCC-related visits to a specialist between the control and intervention group during a period of one year. During a fixed calendar period, questionnaires were not sent to participants due to a logistic error, which lead to data being missing completely at random (MCAR). Although this was the main reason for missing data, some other values were missing as well and missingness was associated with age. Therefore, we considered data to be missing at random (MAR) as well, and applied multiple imputation (MI) using SPSS® 26. With MAR data, MI increases the precision and reduces the level of bias compared to a complete case analysis [27–29]. The imputed data were used for further modelling.

## Cost-utility analysis

The cost-utility was estimated through decision modeling using a patient level health state transition model with a societal perspective. This decision model simulates potential effects on health outcomes and costs that patients would have made over time to estimate the effect of the intervention compared with usual care [19, 30]. The model consisted of several health states between which patients can transition once per cycle. The following health states were included: full recovery from first BCC, new BCC, new cSCC, new melanoma, death due to skin cancer and death due to other causes. Each cycle represented one year and allowed one transition from one state to another, except for death, which functions as an absorbing state. A schematic overview of the model structure can be found in Fig 1.

The study patients' characteristics were used to simulate a cohort of 10,000 hypothetical patients in the model. BCC is a condition with very low mortality rates; therefore, the incremental effects and costs were expected to stabilize after five years. To assess the stability of these incremental outcomes the model-based analysis was performed for a five- and ten-year time horizon. Each health state comes with certain costs and health utilities. Using a societal perspective, costs were accrued in different categories; medical costs, costs for patients and the costs of productivity losses. All costs were presented in euros () and converted to price level 2019 using the Dutch derived consumer price indices [31]. Costs were discounted by 4% and health outcomes by 1.5% according to the Dutch guideline for economic evaluations in healthcare [22, 32]. Disaggregated total deterministic costs were used to determine the impact on each cost category. The model did not include costs for the health states BCC, cSCC and melanoma, because the probabilities of getting one of these diagnoses is equal in both groups and therefore the costs would be expected to be equal. Utility values resulting from the EQ-5D-3L were measured to be able to conduct a cost-utility analysis. However, because the utilities were similar in both groups, the cost-utility analysis was substituted to a cost-minimization analysis. The measured health outcomes were expressed in QALYs and calculated via the scores

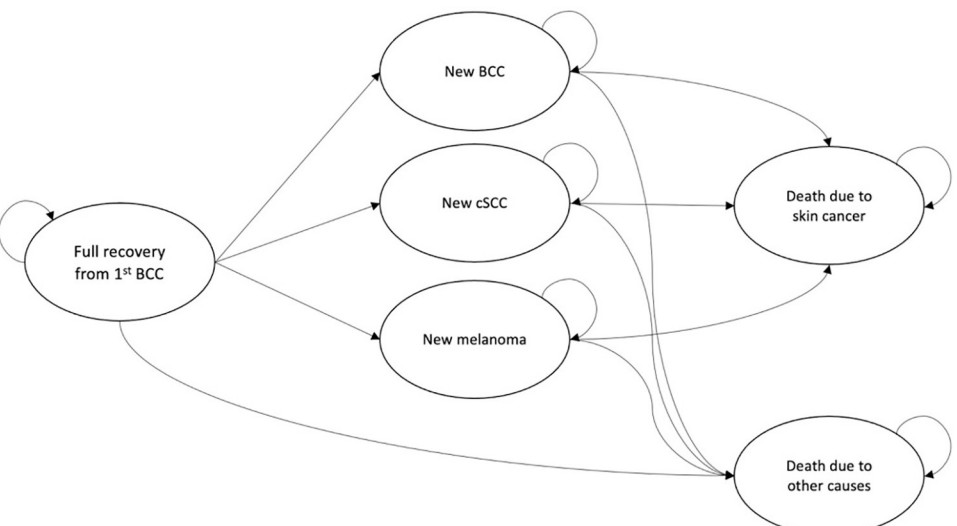

**Fig 1. Schematic overview of the model.** This model focuses on the effects of a personalized discharge letter on first-time BCC patients. Once patients receive a new diagnosis, they no longer meet the criteria of the first group. Expert consensus was reached over the fact that any new diagnosis might influence the effect of the letter, both positively and negatively. To solely model the effect of the letter, patients within the model cannot return to the first state once they have left it.

resulting from the EQ-5D-3L questionnaire. Full descriptions of the included costs, probabilities, utilities and their sources are provided in S2 and S3 Appendices in S1 File. The efficacy results were used to model the number of appointments that were made. For the remaining four years, previous trial data were used to estimate the number of visits per year [33]. The primary outcomes of the model were the total and incremental costs and utilities.

To investigate the impact of the joint parameter uncertainty on the results, a probabilistic analysis (PA) was carried out. The PA shows how variation in the input parameter values affect the outcomes of the mode [34]. The PA was performed with 5,000 Monte Carlo simulations. The Monte Carlo simulation calculated the outcomes of the model by simultaneously drawing random parameter values from previously determined probability distributions which were assigned to each input parameter. The health utilities were defined using a beta distribution, whereas the costs were defined using a gamma distribution. Variation in the number of appointments made was simulated using a Dirichlet distribution [35]. Prices were obtained from the Dutch Costing Manual or aggregated from national data and were therefore free of uncertainty [36]. As a result they were fixed (i.e., without distribution). The software related costs to implement the ability to generate the personalized discharge letter in the electronic health record system were estimated at 5,000 for the first hospital by an IT-specialist. Any additional hospital would have to pay an estimated 1,000 for implementation of the software. To address the uncertainty surrounding this estimate, a scenario analysis was conducted which considered implementation in a single hospital and national implementation. The model was re-run to monitor its effect on the outcomes. The parameter inputs are listed in Table 1. The results of the PA were visualized in the incremental cost-utility plane. Both the model and the PA were performed using Microsoft® Excel 2019 for Mac.

## Budget impact analysis

A BIA was performed to calculate the budgetary impact of implementing the intervention in the Netherlands for a time horizon of five years. The BIA has a societal perspective equal to the CUA and shows the impact for the involved parties [23]. The eligible population for this BIA were low-risk BCC patients diagnosed with skin cancer for the first time. About 48,000 individuals were diagnosed with a BCC in the Netherlands in 2017 [7]. Half of them were considered to be low-risk [15]. With an annual growth of 8% in new cases, the eligible population was calculated for five years [4]. The intervention uptake was defined as 40% in 2021, 50% in 2022, 60% in 2023, 75% in 2024, and 75% in 2025, based on estimates made by dermatologists, an implementation expert, and the results of focus group sessions held with dermatologists [12]. To address the uncertainty surrounding this estimate, scenario analyses with a lower and higher uptake were conducted. These scenarios were also run for local (single hospital) and national implementation.

## Results

### Efficacy

The results of 473 first-time BCC patients were used for this analysis; 278 patients were included as controls and thereafter 195 patients received the intervention. Their characteristics are listed in Table 2.

The number of visits to a specialist in the first year was 1.34 and 1.59 per patient in the intervention group and control group respectively. The distribution of a personalized discharge letter reduced the number of visits by 14.8% (95% CI 0.5% - 29.1%; p = 0.04) after multiple imputation.

**Table 1. Parameter inputs.**

| Parameter | Value | Standard error | Distribution | Source |
|---|---|---|---|---|
| General practitioner appointments | | | | |
| Intervention | | | | |
| 0 | 0.986 | - | Beta | Trial |
| 1 | 0.014 | - | Beta | Trial |
| Control | | | | |
| 0 | 0.978 | - | Beta | Trial |
| 1 | 0.021 | - | Beta | Trial |
| Specialist appointments | | | | |
| Intervention year 1 | | | | |
| 0 | 0.209 | - | Dirichlet | Trial |
| 1 | 0.380 | - | Dirichlet | Trial |
| 2 | 0.266 | - | Dirichlet | Trial |
| 3 | 0.114 | - | Dirichlet | Trial |
| 4 | 0.023 | - | Dirichlet | Trial |
| 5 | 0.006 | - | Dirichlet | Trial |
| 6 | 0.002 | - | Dirichlet | Trial |
| 7–10 | 0.000 | - | Dirichlet | Trial |
| 10 | 0.000 | - | Dirichlet | Trial |
| Control year 1 | | | | |
| 0 | 0.138 | | Dirichlet | Trial |
| 1 | 0.325 | | Dirichlet | Trial |
| 2 | 0.375 | | Dirichlet | Trial |
| 3 | 0.136 | | Dirichlet | Trial |
| 4 | 0.018 | | Dirichlet | Trial |
| 5 | 0.006 | | Dirichlet | Trial |
| 6 | 0.002 | | Dirichlet | Trial |
| 7–10 | 0.000 | | Dirichlet | Trial |
| >10 | 0.000 | | Dirichlet | Trial |
| Control year 2 | | | | |
| 0 | 0.700 | - | Dirichlet | [15] |
| 1 | 0.210 | - | Dirichlet | [15] |
| 2 | 0.040 | - | Dirichlet | [15] |
| 3–5 | 0.040 | - | Dirichlet | [15] |
| 6–10 | 0.010 | - | Dirichlet | [15] |
| >10 | 0.000 | - | Dirichlet | [15] |
| Control year 3 | | | | |
| 0 | 0.820 | - | Dirichlet | [15] |
| 1 | 0.120 | - | Dirichlet | [15] |
| 2 | 0.050 | - | Dirichlet | [15] |
| 3–5 | 0.010 | - | Dirichlet | [15] |
| 6–10 | 0.000 | - | Dirichlet | [15] |
| >10 | 0.000 | - | Dirichlet | [15] |
| Control year 4 | | | | |
| 0 | 0.780 | - | Dirichlet | [15] |
| 1 | 0.160 | - | Dirichlet | [15] |
| 2 | 0.050 | - | Dirichlet | [15] |
| 3–5 | 0.000 | - | Dirichlet | [15] |

(*Continued*)

**Table 1.** (Continued）

| Parameter | Value | Standard error | Distribution | Source |
|---|---|---|---|---|
| 6–10 | 0.000 | - | Dirichlet | [15] |
| >10 | 0.000 | - | Dirichlet | [15] |
| Control year 5 | | | | |
| 0 | 0.940 | - | Dirichlet | [15] |
| 1 | 0.030 | - | Dirichlet | [15] |
| 2 | 0.030 | - | Dirichlet | [15] |
| 3–5 | 0.000 | - | Dirichlet | [15] |
| 6–10 | 0.000 | - | Dirichlet | [15] |
| >10 | 0.000 | - | Dirichlet | [15] |
| Recurrence quality of life | | | | |
| No recurrence | 0.910 | 0.010 | Beta | Trial |
| BCC after BCC | 0.910 | 0.025 | Beta | Trial |
| cSCC after BCC | 0.910 | 0.025 | Beta | Trial |
| Melanoma after BCC | 0.719 | 0.011 | Beta | [37] |
| Recurrence after first BCC | | | | |
| BCC | 0.258 | 0.052[a] | Beta | [38] |
| cSCC | 0.045 | 0.009[a] | Beta | [38] |
| Melanoma | 0.004 | 0.001[a] | Beta | [38] |
| Mortality | | | | |
| BCC | 0.001 | 0.000[a] | Beta | [39] |
| cSCC | 0.021 | 0.004[a] | Beta | [40] |
| Melanoma | 0.071 | 0.014[a] | Beta | [41] |
| General | S3 Appendix in S1 File | - | Fixed | [42] |
| Average productivity loss in hours | | | | |
| Intervention | 0.795 | 0.159[a] | Gamma | Trial |
| Control | 1.606 | 0.321[a] | Gamma | Trial |
| Costs | | | | |
| Follow-up at SP | 117.92 | - | Fixed | [36] |
| Follow-up at GP | 34.95 | - | Fixed | [43] |
| Intervention | 1.61 | 0.322[a] | Gamma | [44] |
| Software | 5,000.00 | - | Fixed | Expert estimate |
| Travel expenses | 2.78 | - | Fixed | [36] |
| Productivity loss male | 39.56 | - | Fixed | [36] |
| Productivity loss female | 32.98 | - | Fixed | [36] |

Abbreviations: BCC, basal cell carcinoma; cSCC, cutaneous squamous cell carcinoma; SP, medical specialist; GP, general practitioner.

[a] ±20% of the deterministic value

## Cost-utility

The costs per patient after the first five years were 348.70 for the intervention and 373.16 for the control group. This resulted in a cost saving of 24.45 per patient. A time horizon of ten years resulted in total costs of 390.37 for the intervention group and 414.79 for the control group. After ten years the expected cost saving would be 24.41 per patient, confirming that effects stabilized after five years. After a period of five years and ten years, the QALYs were 4.12 and 7.25, respectively in both groups. The QALYs in both groups were very similar with differences between the groups being smaller than 0.003 for five years and 0.009 for ten years in favor of the intervention group. The deterministic results of the intervention showed that

**Table 2. Patient characteristics.**

| | Intervention (n = 195) | Control (n = 278) | P-value |
|---|---|---|---|
| **Sex [n (%)]** | | | |
| *Male* | 94 (48.2) | 145 (52.2) | 0.75 |
| **Age (in years)** | | | |
| Mean (SD) | 64.8 (12.7) | 66.4 (11.8) | 0.15 |
| Range | 31–94 | 31–100 | |
| **Education [n (%)]** | | | |
| *Low* | 53 (27.1) | 90 (32.4) | |
| *Medium* | 84 (43.1) | 104 (37.4) | 0.38 |
| *High* | 62 (26.7) | 74 (26.7) | |
| **Self-reported previous actinic keratosis [n (%)]** | | | |
| *Actinic Keratosis* | 7 (3.6) | 4 (1.4) | 0.16 |
| *None* | 169 (86.7) | 250 (89.9) | |
| **VAS score at t = 0** | | | |
| Mean (SD) | 81.45 (11.8) | 79.65 (14.5) | 0.56 |
| **Range** | 30–100 | 19–100 | |

Abbreviations: SD, standard deviation; VAS, visual analogue scale. Means and SDs were compared between using a t-test for independent samples. Significance was set at p < 0.05.

the cost category medical costs accounted to 84.2% of the total costs. The costs for patients made up 2.7% and the remaining 13.1% was made up of productivity losses. The outcomes of the PA showed that 96% of the simulations were in the southern quadrants (Fig 2). This indicates that the intervention is very likely to be cost saving.

A scenario analysis was conducted to measure the effect of uncertainty surrounding the costs of the software during implementation (S4 Appendix in S1 File). When the software

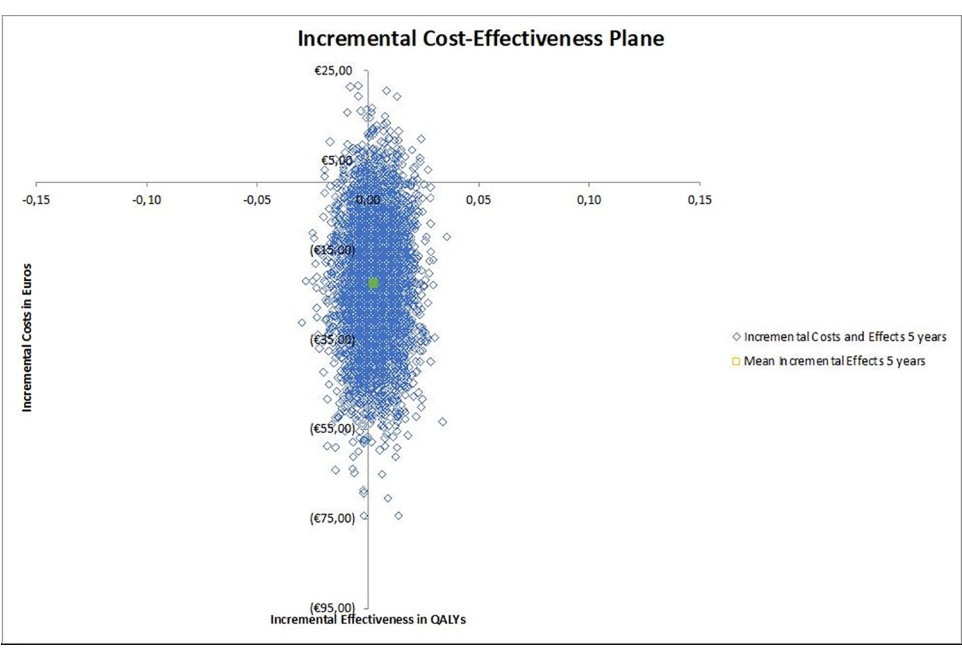

**Fig 2. Incremental cost-utility plane.** Most simulations were in the southern quadrant, indicating that intervention is most likely to be cost saving.

**Table 3. Budget impact analysis of the implementation of the personalized discharge letter.**

| Year | 2021 | 2022 | 2023 | 2024 | 2025 | Total |
|------|------|------|------|------|------|-------|
| **Local implementation (Erasmus MC, Academic Hospital)** | | | | | | |
| Eligible population | 1,260 | 1,361 | 1,470 | 1,587 | 1,714 | 7,392 |
| Expected uptake | 40% | 50% | 60% | 75% | 75% | 75% |
| Patients receiving the intervention | 504 | 680 | 882 | 1,190 | 1286 | 4,542 |
| Budget impact | -11,560 | -15,607 | -20,226 | -27,305 | -29,490 | -104,188 |
| *Lower estimate* | -5,780 | -9,364 | -13,484 | -18,203 | -19,659 | -66,491 |
| *Higher estimate* | -14,451 | -18,728 | -26,968 | -29,126 | -31,456 | -120,728 |
| **National implementation** | | | | | | |
| Eligible population | 30,247 | 32,667 | 35,280 | 38,103 | 41,151 | 177,448 |
| Expected uptake | 40% | 50% | 60% | 75% | 75% | 75% |
| Patients receiving the intervention | 12,099 | 16,333 | 21,168 | 28,577 | 30,863 | 109,040 |
| Budget impact | -295,852 | -399,400 | -517,623 | -698,791 | -754,694 | -2,666,361 |
| *Lower estimate* | -147,926 | -239,640 | -345,082 | -465,861 | -503,129 | -1,701,638 |
| *Higher estimate* | -369,815 | -479,280 | -690,164 | -745,377 | -805,007 | -3,089,644 |

costs were divided over a local setting (one hospital) of 1,000 patients, the cost savings were 22.94 per patient, as mentioned before. If the software costs were divided over the national eligible population of about 31,000 patients, the cost saving was 24.41 per patient (an increase of 6.4%).

## Budget impact analysis

The budget impact of implementing the intervention in the Netherlands would be -2,666,361 over a five-year period. Implementing the intervention in a single hospital resulted in a budget impact of -104,188. The cost savings for each year and scenario are specified in Table 3.

After five years the intervention would result in cost savings on a national scale in the medical category of 2,245,457. The costs for all patients would drop by 71,245. The productivity loss would lower by 349,659. In the local setting the cost savings for the medical category would be 84,079, for the patients 2,500 and for productivity loss 17,608.

A scenario analysis was conducted to determine the impact of a lower and higher uptake (S5 Appendix in S1 File). The total national savings on a lower estimate were 1,701,638. The total savings on a high estimate were 3,089,644. Locally, the lower estimate was -66,491 and the high estimate was -120,728.

## Discussion

To our knowledge, this is the first study evaluating the efficacy, cost-minimization, and budget impact of a personalized discharge letter in dermatology. The introduction of a personalized discharge letter resulted in 14.8% fewer follow-up visits in one year compared to usual care. This would save 24.45 per patient in five years, without affecting the QALYs, while leading to a cost-saving of 2,7 million at the national level in the same period.

The cost category that experienced the highest cost saving were the medical costs, followed by productivity losses and finally the costs for patients. Insurance companies who pay for the medical costs will therefore experience the most (financial) benefits from implementation. The employers of patients, who will have lower productivity losses, are the second largest party who will benefit. Finally, the patients themselves will experience lowered costs.

To implement this intervention, an investment has to be made to develop the required software. After its development, other hospitals could participate as well. When more hospitals join, the number of patients who participate will be higher, resulting in lower intervention costs and an even greater benefit for all involved parties.

There were no data available on the uptake of such an intervention among dermatologists. The dermatologists who participated in the current trial were eager to implement the letter into their routine care. To improve the chances of success, a well-defined implementation plan could increase the uptake [45]. Making the letter available in multiple languages or adding more graphical features can help to include harder to reach populations [46, 47]. It is also important that the letter is easy to create. The less hassle it is to create the letter, the more likely that it will be used in practice. This could be achieved through software that automatically creates the letters and does not require manual adaptations [48]. Prior research has emphasized this; personalized information is rated positively by both patients and professionals and the uptake of the innovation improves when the letter is added to the electronic patient files [49]. Apart from this, personalized information in general has proven to be cost-effective in the long run [50].

During the COVID-19 pandemic, there is an increased need for efficient capacity management. Personalized discharge letters could also be created for other (skin) conditions to reduce low-value follow-up, such as melanoma. The Dutch melanoma guideline states that patients with stage 1A melanoma should receive a single follow-up visit one month after treatment to answer remaining questions, identify potential psychosocial problems and to provide instructions for self-examination [51]. If these patients also receive low-value follow-up, a personalized discharge letter may provide a solution in this situation as well.

A barrier for implementation of this intervention is the effect on dermatologists, as follow-up visits provide revenue. A financial incentive was already identified as barrier for de-adoption in previous research [12]. On the other hand, the reduction of visits lowers the strain on dermatological care and the available time creates space for other, more pressing, consults. Barriers such as these should be considered and taken into the implementation plan, such as increased compensation for more complex skin cancer patients. However, minimizing excessive follow-up could be financially interesting for patients, employers, and insurers. Lowering their expenditures functions as a facilitator for these groups, which could positively influence the implementation process.

## Limitations and strengths

One of the limitations of this study was that patients in the control and intervention group were included in different time periods. During the study period, the first skin cancer guideline was published for GPs which might have altered the clinical practice and therefore the outcome of the intervention [52]. The guideline was published in 2017 and the inclusion period of the intervention group started shortly after its publication. It states that after complete excision and a post-treatment evaluation after 3 months, further monitoring is not necessary. Adoption of guidelines among clinicians is often slow and patient characteristics were very similar in both groups, making it unlikely that it had a significant effect on the outcomes [53]. However, there have not been any changes during the study period in the dermatologist guidelines regarding BCC follow-up. Another limitation was that the effect of the letter has only been monitored for one year. Since patients usually received one or two follow-up appointments, it is possible that the effect of the letter continued beyond the twelve months of monitoring, which would cause an underestimation of the efficacy. Lastly, we did not register if a patient's BCC was incompletely excised, which could increase the number of follow-up visits for those

patients. A strength of the design was that the trial was conducted at different types of dermatological departments (i.e., university medical center, general hospital, independent sector treatment center), which each have their specific target population creating a representative study population, making the results more generalizable to all Dutch BCC patients.

## Conclusions

In conclusion, the personalized discharge letter decreases the amount of low-value follow-up visits among first-time BCC patients. It is a cost-effective strategy and has a positive impact on the healthcare budget. The letter also provides a solution for the patients' need for more and tailored information. Incorporating this intervention in routine BCC care can improve patients' satisfaction with care, helps to decrease the number of unnecessary follow-up visits, and subsequently lowers the costs.

## Supporting information

**S1 File. Contains all the supporting tables and figures.**
(DOCX)

## Author Contributions

**Conceptualization:** Tamar E. C. Nijsten, Esther de Vries, Esther W. de Bekker-Grob.

**Data curation:** Sven van Egmond, Ella D. van Vliet, Yesim Misirli, Rachel S. L. A. Bakkum, Maarten T. Bastiaens, Nicole A. Kukutsch, Albert J. Oosting, Elsemieke I. Plasmeijer, Annik van Rengen, Kees-Peter de Roos.

**Formal analysis:** Sven van Egmond, Ella D. van Vliet, Loes M. Hollestein, Xavier G. L. V. Pouwels, Hendrik Koffijberg, Yesim Misirli, Esther W. de Bekker-Grob.

**Funding acquisition:** Tamar E. C. Nijsten, Esther de Vries, Esther W. de Bekker-Grob.

**Investigation:** Marlies Wakkee, Loes M. Hollestein, Tamar E. C. Nijsten, Esther de Vries, Esther W. de Bekker-Grob.

**Methodology:** Ella D. van Vliet, Marlies Wakkee, Loes M. Hollestein, Xavier G. L. V. Pouwels, Hendrik Koffijberg, Tamar E. C. Nijsten, Esther de Vries, Esther W. de Bekker-Grob.

**Project administration:** Sven van Egmond, Ella D. van Vliet, Xavier G. L. V. Pouwels, Yesim Misirli, Rachel S. L. A. Bakkum, Maarten T. Bastiaens, Nicole A. Kukutsch, Albert J. Oosting, Elsemieke I. Plasmeijer, Annik van Rengen, Kees-Peter de Roos.

**Supervision:** Marlies Wakkee, Loes M. Hollestein, Hendrik Koffijberg, Tamar E. C. Nijsten, Esther de Vries, Esther W. de Bekker-Grob.

**Validation:** Loes M. Hollestein, Xavier G. L. V. Pouwels, Hendrik Koffijberg.

**Writing – original draft:** Sven van Egmond, Ella D. van Vliet.

**Writing – review & editing:** Sven van Egmond, Ella D. van Vliet, Marlies Wakkee, Loes M. Hollestein, Xavier G. L. V. Pouwels, Hendrik Koffijberg, Yesim Misirli, Rachel S. L. A. Bakkum, Maarten T. Bastiaens, Nicole A. Kukutsch, Albert J. Oosting, Elsemieke I. Plasmeijer, Annik van Rengen, Kees-Peter de Roos, Tamar E. C. Nijsten, Esther de Vries, Esther W. de Bekker-Grob.

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
