## [Decision Letter · Decision Letter 0]

30 Sep 2021

PONE-D-21-24499Efficacy, cost-utility, and budget impact of a personalized discharge letter for basal cell carcinoma patients to reduce low-value follow-up carePLOS ONE

Dear Dr. van Egmond,

Thank you for submitting your manuscript to PLOS ONE. After careful consideration, we feel that it has merit but does not fully meet PLOS ONE’s publication criteria as it currently stands. Therefore, we invite you to submit a revised version of the manuscript that addresses the points raised during the review process.

Please find below the issues identified by the peer reviewers and compose the corresponding minor revision of the manuscript. It is acknowledged that the template personalized discharge letter is already included in the Supporting information. During your revision, please also take into consideration the following editorial comments: 1) A key modelling assumption was that the introduction of personalized discharge letter had no effect on clinical outcomes and patient QALYs (i.e., the "low-value" follow-up visits were assumed to have zero clinical value in the model). Hence, it is recommended to discuss how evidence-based this assumption was. Given that no QALY difference was allowed across intervention and control cohorts by the model structure (except for stochastic difference in the probabilistic sensitivity analysis), the developed model is considered to be a cost minimization model, rather than cost-utility model: please consider adjusting the manuscript title accordingly.  2) For better reproducibility, please make an explicit statement about the applied appointment probabilities in Years 2-9 in the intervention cohort in Table S3.2.  3) In line with the comments of Reviewer 1, please include the verbatim recommendations on follow-up visit frequency of the corresponding guidelines for the two cohorts, i.e., before and after the implementation of the personalized discharge letter approach. This information will help readers to interpret any possible interference of guideline revision with the study intervention.  Please submit your revised manuscript by Nov 14 2021 11:59PM. If you will need more time than this to complete your revisions, please reply to this message or contact the journal office at plosone@plos.org. Please include the following items when submitting your revised manuscript:A rebuttal letter that responds to each point raised by the academic editor and reviewer(s). You should upload this letter as a separate file labeled 'Response to Reviewers'.A marked-up copy of your manuscript that highlights changes made to the original version. You should upload this as a separate file labeled 'Revised Manuscript with Track Changes'.An unmarked version of your revised paper without tracked changes. You should upload this as a separate file labeled 'Manuscript'.

We look forward to receiving your revised manuscript.

Kind regards,

János G. Pitter, MD, PhD

Academic Editor

PLOS ONE

Journal Requirements:

2. Thank you for stating the following in the Acknowledgments/ Funding Section of your manuscript: 

This project was funded by Citrienfonds (Dutch Ministry of Health, Welfare and Sport) and VGZ (Health insurance company). The funders were not involved in study design, data collection, data analysis, and manuscript preparation.

This project was funded by Citrienfonds (Dutch Ministry of Health, Welfare and Sport, https://www.citrienfonds.nl/, received by TN) and VGZ (Health insurance company, https://www.vgz.nl/, received by EdV). The funders were not involved in study design, data collection, data analysis, and manuscript preparation.

3. We note that Appendix S1 includes an image of a participant. 

Reviewers' comments:

Reviewer's Responses to Questions

**Comments to the Author**

1. Is the manuscript technically sound, and do the data support the conclusions?

Reviewer #1: Partly

Reviewer #2: Yes

2. Has the statistical analysis been performed appropriately and rigorously? 

Reviewer #1: I Don't Know

Reviewer #2: Yes

3. Have the authors made all data underlying the findings in their manuscript fully available?

Reviewer #1: Yes

Reviewer #2: Yes

4. Is the manuscript presented in an intelligible fashion and written in standard English?

Reviewer #1: Yes

Reviewer #2: Yes

5. Review Comments to the Author

Reviewer #1: Thank you for asking me to review this paper.

This paper looks at the use of a personalised software generated discharge letter to reduce follow up visits following a first BCC.

Prevalence is misused as a term- 80% of Caucasians don’t have a BCC.

They don’t define a low risk BCC- or justify 50% of BCCs in their cohort being low risk.

They should be clear if their group pf BCCS were all completely excised and if so what parameters they use to define that – e.g mohs/ 1mm or more histological clearance at all margins on standard surgical excision with curative intent.

Software related costs are mentioned- they should state this is one way of delivering this, but as with other patient letters, software is not mandatory.

The introduction of new guidelines could have had a significant impact on the behaviour of

and while the authors are right to mention is, it is a major risk that this

led to some or most of the reduction in visits. They should state how follow up guidance changed to be clear on what the impact may have been

This paper demonstrates well that follow up was reduced coinciding with introduction of a personalised discharge letter.

They explore that there is potentially a financial disincentive for clinicians to discharge a patient at one follow up visits. It is common practice in the UK for further visits to simply not be funded and it would be useful to discuss the pros and cons of excessive follow ups, from a patient and clinician and health care economy perspective.

Reviewer #2: Thank you for the nice report. It is nicely written. Correct use of statistics and well explained. The article adds value to current literature.

What is written in the personalized discharge letter? An example in het Supplementary would be usefull. What factors in the letter make it 'personal'/ what are the differences between the letters and based on which characteristics of the patient?

I have no further comments.

6. PLOS authors have the option to publish the peer review history of their article (what does this mean?). If published, this will include your full peer review and any attached files.

---

## [Author Response · Author response to Decision Letter 0]

14 Nov 2021

Reviewer #1: Thank you for asking me to review this paper. This paper looks at the use of a personalised software generated discharge letter to reduce follow up visits following a first BCC.

1) Prevalence is misused as a term- 80% of Caucasians don’t have a BCC.

 The term was indeed misused and has been adjusted accordingly. (p.3, line 76)

2) They don’t define a low risk BCC- or justify 50% of BCCs in their cohort being low risk.

Low-risk BCCs are defined as primary BCC, < 2 cm, located outside the H-zone, with a nodular or superficial subtype according to the Dutch BCC guidelines. This has been added to the manuscript. (p.3, lines 86-87)

Regarding the justification of 50% of BCCs; This number was extrapolated from the previous work of the authors (de Vries et al.), as cited in the manuscript. 

3) They should be clear if their group pf BCCS were all completely excised and if so what parameters they use to define that – e.g mohs/ 1mm or more histological clearance at all margins on standard surgical excision with curative intent.

We indeed do not provide the numbers of completely excised BCCs. This has not been registered for all patients. We acknowledge that this is a limitation of our study and have added it as such to the manuscript. (p.16, lines 354-355)

4) Software related costs are mentioned- they should state this is one way of delivering this, but as with other patient letters, software is not mandatory.

By software costs we intended the costs needed to implement the feature in the current electronic health record system the hospital already uses. We have clarified this in p.7, lines 198-199.

5) The introduction of new guidelines could have had a significant impact on the behaviour of

and while the authors are right to mention is, it is a major risk that this led to some or most of the reduction in visits. They should state how follow up guidance changed to be clear on what the impact may have been

We agree with the reviewer that clarifying this is valuable for the interpretation of the discussion. We have clarified in the discussion that this is the first skin cancer guideline for GPs (p.15, line 340) and that the guidelines states that after complete excision and a post-treatment evaluation after 3 months, further monitoring is not necessary (p.15, lines 343-344). However, we would like to emphasize that there have not been any changes in the dermatologist guidelines regarding BCC follow-up care, and the dermatologist is in control of the patients’ follow-up schedule (p16, lines 349-350)

6) This paper demonstrates well that follow up was reduced coinciding with introduction of a personalised discharge letter. They explore that there is potentially a financial disincentive for clinicians to discharge a patient at one follow up visits. It is common practice in the UK for further visits to simply not be funded and it would be useful to discuss the pros and cons of excessive follow ups, from a patient and clinician and health care economy perspective.

We agree with the reviewer that a further exploration on the effect of low-value care is an interesting direction. We have chosen to focus on the possible implementation effects of this particular intervention on all affected parties, ranging from employers to clinicians and insurers. We have added an extra perspective to the manuscript (p.15, line 335-337). 

Reviewer #2: Thank you for the nice report. It is nicely written. Correct use of statistics and well explained. The article adds value to current literature. 

1) What is written in the personalized discharge letter? An example in het Supplementary would be useful. What factors in the letter make it 'personal'/ what are the differences between the letters and based on which characteristics of the patient?

I have no further comments.

We are sorry to notice that there might have been an administrative issue during the submittal process. An example of the personalized letter was already added in the supplementary. We hope that this will find you well in the current rebuttal.

Editorial comments

1) A key modelling assumption was that the introduction of personalized discharge letter had no effect on clinical outcomes and patient QALYs (i.e., the "low-value" follow-up visits were assumed to have zero clinical value in the model). Hence, it is recommended to discuss how evidence-based this assumption was. Given that no QALY difference was allowed across intervention and control cohorts by the model structure (except for stochastic difference in the probabilistic sensitivity analysis), the developed model is considered to be a cost minimization model, rather than cost-utility model: please consider adjusting the manuscript title accordingly. 

The effect of the personalized letter on the utility values was measured in the current study by administrating the EQ-5D-3L questionnaire. We agree with the editor that by having similar utility values in each group, the analysis is in fact a cost-minimization analysis. Extra clarification has been added to the manuscript (p. 7, line 180-182). We have also changed the title to “Efficacy, cost-minimization, and budget impact of a personalized discharge letter for basal cell carcinoma patients to reduce low-value follow-up care”

2) For better reproducibility, please make an explicit statement about the applied appointment probabilities in Years 2-9 in the intervention cohort in Table S3.2. 

We have improved the readability by starting with the control group and providing extra specification about the probabilities in the supplementary material. 

3) In line with the comments of Reviewer 1, please include the verbatim recommendations on follow-up visit frequency of the corresponding guidelines for the two cohorts, i.e., before and after the implementation of the personalized discharge letter approach. This information will help readers to interpret any possible interference of guideline revision with the study intervention.

We agree with both the reviewer and editor that further specification was needed regarding the possible guideline interference. This has been adjusted accordingly as stated in response to reviewer 1, comment 5.

---

## [Editor Report · Decision Letter 1]

22 Nov 2021

Efficacy, cost-minimization, and budget impact of a personalized discharge letter for basal cell carcinoma patients to reduce low-value follow-up care

PONE-D-21-24499R1

Dear Dr. van Egmond,

We’re pleased to inform you that your manuscript has been judged scientifically suitable for publication and will be formally accepted for publication once it meets all outstanding technical requirements.

Kind regards,

János G. Pitter, MD, PhD

Academic Editor

PLOS ONE
---

## [Editor Report · Acceptance letter]

12 Jan 2022

PONE-D-21-24499R1 

Efficacy, cost-minimization, and budget impact of a personalized discharge letter for basal cell carcinoma patients to reduce low-value follow-up care 

Dear Dr. van Egmond:

I'm pleased to inform you that your manuscript has been deemed suitable for publication in PLOS ONE. Congratulations! Your manuscript is now with our production department. 

Kind regards, 

on behalf of

Dr. János G. Pitter 

Academic Editor

PLOS ONE